# Supplementary Pharmacotherapy for the Behavioral Abnormalities Caused by Stressors in Humans, Focused on Post-Traumatic Stress Disorder (PTSD)

**DOI:** 10.3390/jcm12041680

**Published:** 2023-02-20

**Authors:** Jeffrey Fessel

**Affiliations:** Department of Medicine, University of California, 2069 Filbert Street, San Francisco, CA 94123, USA; jeffreyfessel@gmail.com

**Keywords:** stressors, behavioral abnormality, PTSD, traumatic brain injury, chronic traumatic encephalopathy, treating changed brain cell types, pharmacotherapy, two-drug combination, three-drug combination

## Abstract

Used as a supplement to psychotherapy, pharmacotherapy that addresses all of the known metabolic and genetic contributions to the pathogenesis of psychiatric conditions caused by stressors would require an inordinate number of drugs. Far simpler is to address the abnormalities caused by those metabolic and genetic changes in the cell types of the brain that mediate the behavioral abnormality. Relevant data regarding the changed brain cell types are described in this article and are derived from subjects with the paradigmatic behavioral abnormality of PTSD and from subjects with traumatic brain injury or chronic traumatic encephalopathy. If this analysis is correct, then therapy is required that benefits all of the affected brain cell types; those are astrocytes, oligodendrocytes, synapses and neurons, endothelial cells, and microglia (the pro-inflammatory (M1) subtype requires switching to the anti-inflammatory (M2) subtype). Combinations are advocated using several drugs, erythropoietin, fluoxetine, lithium, and pioglitazone, that benefit all of the five cell types, and that should be used to form a two-drug combination, suggested as pioglitazone with either fluoxetine or lithium. Clemastine, fingolimod, and memantine benefit four of the cell types, and one chosen from those could be added to the two-drug combination to form a three-drug combination. Using low doses of chosen drugs will limit both toxicity and drug-drug interactions. A clinical trial is required to validate both the advocated concept and the choice of drugs.

## 1. Introduction

Stressors are widely recognized as causing such behavioral abnormalities as excessive irritability, anger, impatience, fear, depression, and anhedonia. Although psychotherapy would be the usual approach to treatment, it is not always successful; yet, there is no generally-approved pharmacotherapy for use as a supplement to psychotherapy. The myriad genetic and metabolic abnormalities that are either associated with or result from stress would require a suffocating, impractical number of drugs for their reversal. However, the task of formulating pharmacotherapy for stress-related behavioral abnormalities may be simplified if one considers that any psychiatric condition ultimately must reflect changes in the function within the brain of one or more of its cell types, i.e., astrocytes, oligodendrocytes, neurons, endothelial cells, and microglia [1], some of which (neurons and astrocytes) also affect the hypothalamus-pituitary-adrenal axis (HPA) [2]. In another article, the formulation of pharmacotherapy in general for patients with major psychiatric conditions is discussed [1]; here, the focus is on those patients whose disturbed behavior results from stressors. Post-Traumatic Stress Disorder (PTSD) is a convenient surrogate for approaching treatment for the behavioral abnormalities caused in humans by stressors because PTSD has been extensively studied so that its underlying molecular biology is known and provides a rational approach to pharmacotherapy that may provide benefit as a supplement psychotherapy. In addition to PTSD, there are useful data in humans with chronic traumatic encephalopathy (CTE) or traumatic brain injury (TBI) that may be taken into account. Based upon data obtained in humans with PTSD, CTE, and TBI, this article will examine the formulation of pharmacotherapy that may be useful for those patients with aberrant behavior caused by stressors and whose response to psychotherapy is inadequate. After a brief summary of the multiple metabolic and genetic factors that participate in the pathogenesis of PTSD, this article will describe the roles of brain cell types. Results from rodent studies are avoided for reasons discussed in another article, but when mentioned, they will have human correlates [1].

## 2. Metabolic and Genetic Changes in PTSD, TBI and CTE

Peripheral and central levels of corticotropin-releasing hormone, which represents the initiating step in the activation of the hypothalamic/pituitary/adrenal (HPA) axis, are elevated, and enhanced glucocorticoid negative feedback on the HPA axis is consistently reported; as a result, individuals with PTSD have elevated levels of the glucocorticoid receptor (GR), and enhanced glucocorticoid sensitivity [3]. The FK506 binding protein 5 (FKBP5) is a cochaperone of the steroid receptor complex, inhibits GR ligand binding and nuclear translocation of GRs, and thus regulates the sensitivity of the GR. FKBP5 is decreased in PTSD. Binder et al. found that child abuse and other childhood traumas predicted the level of adult PTSD symptomatology, and four SNPs in the FKBP5 locus significantly interacted with the severity of childhood trauma to predict the level of adult PTSD symptoms [4].

PTSD also has an associated, significantly elevated risk for metabolic syndrome and its individual components, i.e., obesity, insulin resistance and diabetes, hypertension, and dyslipidemia.

Swanberg et al. reviewed 32 studies using proton magnetic resonance spectroscopy (^1^H MRS) in subjects with PTSD, which showed the hippocampus and anterior cingulate cortex as having significant decreases in N-acetyl aspartate (NAA); since NAA is the second most abundant amino acid in the brain and is involved in protein synthesis, its decrement in PTSD is deleterious [5].

Multiple reports describe a genetic background for PTSD. That may be via several pathways, including the N-methyl-D-aspartate receptor (NMDAR), the dopaminergic pathway, the serotoninergic system, the adrenocorticotropic axis, and pathways affecting the number or functions of brain cell types. Because of these multiple effects, the overall significance of genetic participation in the pathogenesis of PTSD is difficult to summarize. For a comprehensive review, see Ref. [6]; some illustrative examples follow.

The NMDAR ligates glutamate, which is an excitatory neurotransmitter that, in excess, causes neurotoxicity. This receptor is composed of several subunits, and it has as a co-receptor a glycine binding site that modulates the permeability of the NMDAR. The gene GRIN3B expresses the NMDAR subunit 3B. Lori et al. studied 377 patients who had been seen in an Emergency Department immediately after their exposure to trauma which, for 70.3%, was a road accident [7]. During 1, 3, 6, and 12 months after the trauma, blood levels of the mRNA transcribed from GRIN3B were increased in those who developed a future, non-remitting PTSD as compared with levels seen in those who did not develop PTSD. GRIN3B is found, particularly in the neocortex and hippocampus.

Considering the dopaminergic pathway, the A1 allele of the type-2 dopaminergic receptor is associated with comorbidities between PTSD and anxiety, social change and depression [8].

Regarding noradrenergic neuromodulation, there is an interaction between the polymorphism of GABRA2 and the occurrence of PTSD; and there was an association between the number of traumatic events and the V158M allele of the gene coding for catecholamine-o-methyltransferase [8].

With respect to the serotonergic pathway, several studies have found that a high risk of developing PTSD is associated with a specific polymorphism in the promoter region of the serotonin transporter gene SLC6A4 [6].

As a final example, the V66M mutation in the BDNF gene was seen in 33.3% of individuals with PTSD as compared to 17.5% in non-PTSD controls [6]; the V66M mutation affects localization of brain-derived neurotrophic factor (BDNF) and thus affects neuronal cell function [9].

A notable difficulty is that the results in published studies are often conflicting, exemplified by reports from Logue et al., who analyzed blood samples for differentially expressed genes from 115 cases of PTSD and 28 controls, using 10,264 probes for genes or gene transcripts [10]. They found 41 genes that were differentially expressed by the PTSD cases; they then used two replication cohorts to reanalyze for those 41 and now could demonstrate only seven as being significantly associated (*p* < 0.05): but of those seven, only one survived correction for multiple testing. That gene was ATP6AP1L, which is not one of three (BDNF, NA3C1, TXNRD1) that Logue et al. indicated as having been reported by other investigators. Six years later, Logue et al. reported further studies, this time involving 73 cases of PTSD alone, 32 with PTSD plus depression, 32 with depression alone, and 44 controls [11]. They performed 190,346 single-gene analyses. After strict correction for multiple testing, they identified 21 genes differentially expressed by PTSD. Strikingly, ATP6AP1L was not among the 21. Those 21 included three down-regulated genes for oligodendrocytes (MBP, MOBP, ERMN) and two dysregulated genes for microglia. They used two different methods to find correlations of cell types with PTSD: the first method showed high correlations (r > 0.90) for oligodendrocytes and astrocytes; the second method showed zero correlations. It is interesting that the gene for endothelin 1 was up-regulated in the dorsolateral (dl) prefrontal cortex (PFC) of PTSD cases because endothelin 1 causes vasoconstriction, which would be deleterious but it has also been reported to correlate positively, which would be beneficial, with numbers of oligodendrocyte precursor cells (OPC) in humans with strokes and leukoencephalopathy [12].

In sum, there is poor reproducibility of the multiple genetic findings in PTSD, but even were therapies available to modify the effects of those genes, that would require too many drugs and would be impractical. As by pointed out by Willsey et al., neuronal circuits exhibit emergent behavior that reflects the collective properties of multiple cell types [13]. A simpler and more available solution to the problem of pharmacotherapy for PTSD is to address whatever effect these genes exert on the changes seen in brain cell types of persons with PTSD.

## 3. Changes in Brain Cell Types in PTSD, CTE, and TBI

The processes of astrocytes wrap around and affect axons and cerebral microvessels, and they also supply growth factors for oligodendrocytes that promote the proliferation of OPCs. Thinner myelin ensues when contact between astrocytes and oligodendrocytes is lost because myelination derives from oligodendrocytes [14]. There are several ways by which astrocytes are crucial for neurons: (1) they control glutamate clearance via the excitatory amino acid transporter (EEAT) and prevent excitatory neurotoxicity; (2) they synthesize and release BDNF; (3) they control ion (K^+^, Ca^2+^, Cl^−^) homeostasis in the neuropil; (4) they produce a Ca^2+^ binding protein, S100B, which controls neuron firing [15]; (5) via the water channel aquaporin-4 (AQP4) in the end feet of their processes, astrocytes control water transport in the brain and, therefore, entry of drugs into the brain. Astrocytes also produce vascular-derived growth factor (VEGF), through which they benefit cerebral microcirculation [16].

MAO-B is a monoamine oxidase-B enzyme that is primarily located in astrocytes. Gill et al. used positron emission tomography with the MAO-B probe [^11^C]SL25.1188 to assess astrocyte levels, and coincidentally of MAO-B, and found both of them decreased in subjects with PTSD as compared with controls [17].

Wu et al. found that single, prolonged stress (SPS) caused prolonged extinction of contextual fear memory that correlated with increased hippocampal autophagy; and also found that preventing autophagy by astrocytes promoted the extinction of contextual fear memory, which for PTSD would be beneficial [18]. Lei et al. used SPS in rats as a model for PTSD and also saw that it increased autophagy by astrocytes [19], which provides benefits by clearing disease-related proteins in astrocytes and maintaining astrocyte function [20]. After unilateral TBI, an analog of PTSD, activated astrocytes were seen not only ipsilaterally but also in the contralateral neocortex and cingulate cortex (CC), which is important insofar as it shows that a localized TBI may cause a diffuse brain reaction [21].

Thus, data show that astrocytes have substantial importance in PTSD, which should benefit from inducing their increase.

### 3.1. Oligodendrocytes

While under optimal conditions, oligodendrocytes use glycolysis, under metabolic stress, as in PTSD, they have reduced glycolysis and impaired function [22]. Among seven genes identified by Wingo et al., that contribute to the pathogenesis of PTSD, three were expressed by oligodendrocyte precursor cells (OPC) [23]. The fibroblast growth factor 17 (Fgf17) is involved in this: Fgf17 infusion induced OPC proliferation [24]; its receptor is FGFR3, and FGFR3-deficient mice had reduced numbers of differentiated oligodendrocytes in the forebrain, cerebellum, hindbrain, and spinal cord and, in parallel, myelination was delayed [25]. Further, the knock-out of the *Fgf17* gene in mice created deficits in the results of tests for social recognition [26], which would predispose to PTSD because social recognition protects against the development of PTSD [27]. Another linkage is because the Fgf17 gene controls the size of the dorsal prefrontal cortex (dlPFC), which is crucially relevant to behaviors that are seen in PTSD [28]. In brief, the data show the relationship between PTSD, oligodendrocytes, and FGF17.

From eight individuals with neuropathologically confirmed CTE and eight age- and sex-matched controls, Chancellor et al. studied postmortem white matter from the dlPFC [29]. Using the marker genes PLP1, MBP, and CLDN11 to identify mature oligodendrocytes and VCAN and PDGFRα to identify OPC, they found that the total number of oligodendrocyte lineage cells in the CTE white matter was decreased by 41%. Those were chronic cases; acute cases have shown loss of mature oligodendrocytes but a reactive increase in their precursors. Thus, Flygt et al. used brain tissue from ten patients with severe, acute TBI and from autopsies of five age-matched controls; the TBI brains had increased apoptosis of oligodendrocytes (*p* < 0.05) and a striking, inverse correlation with the time since injury (r = −0.8, *p* < 0.05) [30]. Another report showed that TBI brains had reduced myelin basic protein (MBP) [31]. Lotocki et al. confirmed this by showing that TBI in rats led to a reduced number and function of oligodendrocytes [32]. One may infer that persistently reduced axonal myelin due to TBI must be due to a defective response from myelinating oligodendrocytes since oligodendrocyte formation leading to myelination is expected during brain repair [33]. Illustrating this, the cortical brain levels of MBP decreased initially in 157 persons with TBI, then gradually increased during survival [34]. A reasonable interpretation is that the trauma caused initial axonal damage and myelin loss and that oligodendrocyte formation during the period of recovery caused myelination and increased MBP.

Changes in axonal myelin and changes in oligodendrocytes are concurrent; however, increased oligodendrocyte formation is expected during brain repair, so one may infer that persistently reduced axonal myelin due to TBI must be due to a defective response from myelinating oligodendrocytes [33].

Finally, it is worth noting that, for various reasons, oligodendrocytes in different brain regions may have different properties, as discussed by Power et al. in an interesting and relevant article [35].

### 3.2. Neurons

Imaging studies in PTSD show lowered volumes in various regions of the brain that, in part, reflects decrements in either neuronal somata or their processes. In PTSD, volume reduction in the hippocampus and ventromedial (vm) PFC may reflect a reduced cortical capacity to inhibit the fear and negative emotional responses seen in that condition. The Enigma consortium used data for 68 cortical regions across both hemispheres from 1379 PTSD patients and 2192 controls; in prefrontal regulatory regions of PTSD, they saw reduced volumes in orbitofrontal gyri (OFG), the superior temporal gyrus (STG), and the right insular, lingual and superior parietal gyri (all *p* values < 0.039) and those were negatively correlated with PTSD severity [36]. Studies reviewed by Karl et al. confirmed that PTSD is associated with abnormalities in multiple fronto-limbic structures and also noted that hippocampal volumetric differences co-vary with PTSD severity [37]. Hayes et al. reviewed data from 26 studies that, among other findings, showed robust hyperactivity in the dorsal anterior cingulate cortex (aCC) and that hypo-activity in the vmPFC was associated with hyperactivity in the amygdala [38]. Kasai et al. studied monozygotic twins having various combinations of combat exposure in Vietnam and the presence or absence of PTSD [39]. Compared to the combat-exposed twins without PTSD, the combat-exposed monozygotic twins with PTSD showed significant reductions of gray matter density in the right hippocampus, pregenual aCC, and bilateral insulae, suggesting that the gray matter reduction in this region represents an acquired finding in PTSD that is consistent with stress-induced loss, and that the pathogenesis of PTSD involves both genetic factors and environmental trauma.

Girgenti et al. reported differential gene expression in four PFC subregions from postmortem tissue of 52 people with PTSD and 46 matched, neurotypical comparison subjects [40]. Integration of this data set with genotype data from a large PTSD genome-wide association study identified down-regulation of the interneuron synaptic gene ELFN1 as conferring a significant liability for PTSD. ELFN1 recruits presynaptic GRM7 (metabotropic glutamate receptor 7) and GRIK2 (glutamate receptor kainate 2) [41]; both of these receptors are expressed by OPC [42]; thus, reduced ELFN1 would interfere with OPC functions and would impact myelinating, mature oligodendrocytes.

Results from the many imaging studies in PTSD showed marked variation that probably has numerous causes: genetic differences; the nature of the provoking trauma, e.g., childhood neglect and abuse, torture, and war combat; the duration of symptoms; and the severity of symptoms.

### 3.3. Endothelial Cells (ECs)

ECs have prime importance because they form the cerebral microvasculature that supplies oxygen and nutrients to all other brain cells. Logue et al. reported down-regulation of ECs in the ventromedial (vm) PFC, which would adversely affect cerebral microcirculation [11]. EC precursor cells in the brain increased after TBI, and microvessel density in the injured brain increased >4-fold [43]. ECs contain gene transcripts called erythropoietin-producing hepatocellular (Eph) that induce apoptosis; Assis-Nascimento et al. created Eph^−/−^ mice and, using them, demonstrated that after TBI, EphB3 induced decreased endothelial cell survival and, thus, decreased density of blood vessels, and increased blood-brain barrier (BBB) permeability [44].

Clearly, any treatment that dilates cerebral microvessels would be beneficial. Peripheral vasodilatation can be measured easily by ultrasound, but the dilatation of cerebral microvessels cannot be so-assessed. Therefore, investigators turned to the renin-angiotensin system. Renin converts angiotensinogen to angiotensin-1 (AT-1), which is physiologically inactive but becomes converted to angiotensin-2 (AT-2) by one of the angiotensin-converting enzymes (ACE). AT-2 is the major effector peptide of the renin-angiotensin system and causes vasoconstriction by very complex mechanisms (see Haspula and Clark [45]). A highly-simplified description of those mechanisms is that AT-2 stimulates the adrenal release of vasoconstrictive catecholamines, and in the brain, it stimulates the release of vasopressin [which is antidiuretic hormone (ADH)] by the posterior pituitary. The vasopressin produces increased plasma volume, but since another effect of AT-2 is to decrease the sensitivity of the baroreceptor reflex, there is inadequate vasodilation in response to the increased plasma volume, and this produces a local rise in pressure. The antihypertensive drug, losartan, is an AT1 receptor antagonist, so Seligowski et al. examined the medications used by a large hypertensive population. Among the 397 persons using losartan for hypertension, the occurrence of PTSD was reduced by 6.9% in users versus 9.1% in non-users of losartan (*p* < 0.001) [46]. Concordant with this, Pacak et al. showed that exposure of animals to the stress of immobilization markedly and rapidly increased rates of synthesis, release, and metabolism of norepinephrine in the hypothalamus and amygdala [47]. Nevertheless, a placebo-controlled, randomized study in 149 patients with PTSD, reported by Stein et al., showed no benefit from using losartan at a maximum tolerated dosage for ten weeks [48].

### 3.4. Microglia

It is important to note that differing results for microglia may be seen in different brain regions; thus, Li et al. saw the numbers of microglia increased in the hippocampus and PFC but not in the amygdala [49]. While there are few studies in humans that are relevant to the role of microglia in PTSD’s many studies in rodents, reviewed in detail by Enomoto and Kato, affirm that activation of microglia is important for the pathogenesis of PTSD [50]. In humans with PTSD, the only relevant studies are those involving PET scans using the translocator protein (TSPO) and those involving the glucocorticoid (GC) signaling pathway; but because TSPO is expressed by both microglia and astrocytes, and that expression varies according to the time following the event causing a brain reaction, the interpretation of PET studies in humans using TSPO ligands, is difficult (see Ref. [51]). That leaves investigations of the GC signaling pathway to show microglial participation.

### 3.5. The Overall Changes in Brain Cell Types in PTSD, CTE and TBI

In brief, the brain cell types that are changed in PTSD are decreases in astrocytes, oligodendrocytes, neurons and synapses, and endothelial cells, and increases in microglia.

### 3.6. The Hypothalamus-Pituitary-Adrenal (HPA) Pathway in PTSD

The hypothalamus-pituitary-adrenal (HPA) pathway in PTSD has been extensively investigated because stress involves activations of the adrenal gland. For a complete discussion, the reader is referred to reviews by Herman et al., Dunlop and Wong, and Szeszko et al. [52,53,54]. Several brain pathways modulate HPA axis activity. The paraventricular nucleus (PVN) of the hypothalamus is one of the most important autonomic control centers in the brain and has important hormonal receptors, including those for mineralocorticoids, glucocorticoids, and thyroid; those receptors regulate the pituitary and adrenal production of their relevant hormones. The PVN is influenced by signaling molecules (orexins, adiponectin, glutamate, and GABA) but, relevant in the present context, also by neuronal afferents from multiple sources, including the brainstem (the nucleus tractus solitarius and the nucleus of the vagus nerve), PFC, hippocampus, and amygdala, and from within the hypothalamus itself. In the PVN, there are neurons that cause ACTH production by the pituitary; the hippocampus and PFC inhibit them, and the amygdala and brain stem neurons stimulate them; in addition, glucocorticoids exert negative feedback control of the HPA axis and, as shown below, that control is disturbed in PTSD. Changes in these neural circuits have a direct link to the development of PTSD. However, addressed here is the likelihood that involvement of the HPA axis occurs primarily from the antecedent effect of functional changes in the brain cells that are the present topic: clearly, neuronal circuits control the PVN, and their importance is emphasized by studies showing that inactivation of the basolateral amygdala during stress prevented the expected increase in glucocorticoid receptors in the PFC [55], and it also prevented, a reduction of astrocyte numbers that otherwise would occur during stress [56]. It may be concluded that the disturbed HPA pathway in PTSD results from changes in neuronal cells and neuronal tracts that, in turn, result from impaired function of myelinating oligodendrocytes. Besides, with respect to PTSD, chronic stress reduced the length and volume of astrocytic processes by 40.6% and 56%, respectively, and the number of their branch points was reduced by 57.8% [57], and shown above, cerebral endothelial cells are down-regulated, and activated microglia are not inhibited. All evidence shows that pharmacotherapy for PTSD should be directed to brain cell types.

Since both hydrocortisone and dexamethasone cause profound inhibition of microglial actions, their deficient action in PTSD would allow undisturbed effects of microglia [58,59]. Brain cortisol can be assessed by levels of 11β-hydroxysteroid dehydrogenase type 1 (11β-HSD1), which catalyzes the conversion of inert 11-dehydrocorticosterone to active corticosterone; higher levels of 11β-HSD reflect higher brain cortisol and thus, correlate with enhanced suppression of the hypothalamus-pituitary-adrenal (HPA) axis. Observations in PTSD consistently show enhanced sensitivity of the HPA axis negative feedback loop, such that greater cortical and limbic 11β-HSD1 expression is associated with HPA axis suppression. Bhatt et al. found that greater overall PTSD severity was associated with lower pre-frontal-limbic 11β-HSD1 availability, i.e., lower active corticosterone, which would result in reduced suppression of activated microglia [60]. Van Zuiden et al. noted that vulnerability to PTSD includes various dysregulations of the glucocorticoid signaling cascade, including low FKB5 levels in peripheral blood mononuclear cells (PBMC) prior to trauma and single nucleotide polymorphisms (SNP) in genes associated with glucocorticoid signaling such as FKBP5 [61]. Further, a meta-analysis involving 47 studies and 6008 subjects showed that those with PTSD had a lower daily cortisol output [62]. The receptor for the FK506 binding protein (FKBP5) is a co-receptor for the glucocorticoid receptor (GR); its increased levels would cause resistance to glucocorticoids which is seen in PTSD [61]. Thus, all of the factors mentioned in this paragraph show an overall decrease in the availability in the brain of glucocorticoids which associate with less inhibition of activated microglia and greater vulnerability to PTSD.

### 3.7. Available Drugs That Address the Changes in Brain Cell-Types That Underpin the Pathogenesis of PTSD, CTE and TBI

Fortunately, there are several drugs that correct the changes in the affected brain cell types and from which to choose a rational formulation of treatment. From the following list, it is recommended to choose combinations of drugs, using them in reduced dosages, that will minimize the likelihood of adverse events, including those from drug-drug interactions.

Highly abbreviated descriptions for likely therapeutic effects are provided here, showing either an increase that the drugs induce in the number or activation of astrocytes, oligodendrocytes, synapses and neurons, and endothelial cells; or a decrease in the number or activation of microglia.

Clemastine decreased loss of astrocytes [63]; increased postsynaptic proteins [63], muscarinic receptor, synapsin 1 and Homer 1, and improved oligodendrocyte survival/function [64,65]; it enhanced myelin repair [66] and neuronal function [67]; it enhanced myelination in the PFC [68,69]; it also enhanced visual function [65,70]. Clemastine provided mitochondrial protection [71,72]; and suppressed microglial M1 activation [73].

Erythropoietin (EPO) increased the differentiation of neuronal stem cells (NSC) into astrocytes [74], increased oligodendrocytes [74], and neurons and dendritic spines [75] via increased production of brain-derived neurotrophic factor (BDNF) and its receptor [76]; and by enhancing differentiation of NSC [77]. EPO improved the function of synapses [78,79,80], endothelial cells [81,82] and microglia [83].

Fingolimod, an agonist of sphingomyelin phosphate (S1P), produces cell proliferation [84]. In astrocytes, fingolimod activated neurotrophic genes [85] and reduced the formation of ceramide that causes apoptosis [86,87]. It activated myelinating oligodendrocytes [88], increased OPCs, myelination, and neurological function [89], and ameliorated brain demyelination [90]. It enriched synaptic genes [91], prevented synaptic toxicity [92], and reversed synaptic hypersensitivities. It produced myelin in the demyelinated brain [93] and prevented neural death from NMDA [94]. In multiple sclerosis, it improved axonal and myelin integrity [95] and, in mice, prevented demyelination [96]. It increased dendritic spines [97,98], reduced neuronal death from ROS [99], and improved mitochondrial production of ATP [100]. In microglia, fingolimod shifted M1 polarization toward M2 [101,102].

Fluoxetine induced increases in astrocytes [103], oligodendrocytes [104,105,106], neurons [104,105,106], and endothelial cells [107] and decreased microglial activation [106,108,109].

Fluoxetine activates astrocytes to produce BDNF and VEGF [16,103,110] and promotes the clearance of astrocytes with damaged mitochondria [111]. It up-regulated OPC and oligodendrocyte markers [104] and reduced oligodendrocyte senescence [112]. Fluoxetine increased neurogenesis [104,113], neuronal circuits [114], and spatial learning [115]. Neurons deprived of glucose and oxygen had increased survival with fluoxetine [116]. It increased vasodilatation via differentiation and proliferation of endothelial cells and decreased arteriolar tone, and attenuated EC death and disruption of the BBB [107,109,117]. In microglia, it attenuated NADPH oxidase activation, production of ROS and reactive nitrogen species, and down-regulated M1 and up-regulated M2 activation [108,118,119].

Several clinical trials in patients with PTSD have shown the therapeutic benefits of fluoxetine. Martenyi et al. reported a double-blind, randomized, placebo-controlled study conducted primarily in war-torn countries [120]. Among its 301 participants, 48% were exposed to a combat-related traumatic episode. Patients were randomly assigned to 12 weeks of acute treatment with fluoxetine, 20 to 80 mg/day (*n* = 226), or placebo (*n* = 75). Compared with placebo, fluoxetine was associated with a greater improvement from baseline in the primary efficacy measurement, which was the Treatment Outcome PTSD rating scale (*p* = 0.006), and also in significantly greater improvements of the scores in the Clinician-Administered PTSD Scale (CAPS), Clinical Global Impressions (CGI)-Severity of Illness scale, the CGI-Improvement scale, the Montgomery-Asberg Depression Rating Scale, and the Hamilton Rating Scale for Anxiety. The mean fluoxetine dose at the endpoint was 57 mg. There were no clinically significant safety differences. In a subsequent article, Martenyi and Soldatenkova reported on the efficacy of maintenance treatment with fluoxetine as compared with a placebo in 144 veterans of the war in Yugoslavia who had been diagnosed with PTSD [121]. In addition to the rating scales mentioned above, this trial used the Davidson Trauma Scale, which is a standard, clinical global rating scale to which are added separate measures for the three main PTSD symptom clusters (intrusion, avoidance/numbing, and hyperarousal). The risk of relapse in the placebo arm was significantly greater than in the fluoxetine arm (*p* = 0.048). Fluoxetine was well tolerated at a mean daily dose of 65 mg.

Smaller studies have been reported. Connor et al. randomized a civilian sample of PTSD subjects, 27 to receive fluoxetine and 27 to receive a placebo, for 12 weeks [122]. If PTSD symptoms had been present for <6 years, the response rate from fluoxetine was 93% versus 42% from placebo (*p* = 0.008). Two open-label trials without placebo controls have reported successful results from fluoxetine [123,124]. On the contrary, a small trial gave negative results: Hertzberg et al. enrolled twelve male veterans with PTSD in a 12 week double-blind evaluation of fluoxetine and placebo; the mean fluoxetine dose at the endpoint was 48 mg/day [125]. Only one of the six patients taking fluoxetine responded versus two of the six using a placebo.

Lithium doubled astrocytic numbers and their VEGF secretion [126]. Oligodendrocyte expression of PLP and MBP increased, improving synaptic and neuronal function [127,128,129,130,131]. By negating activation of GSK-3β, synaptic expression of PSD-95 and gephyrin were enhanced. Lithium promoted neurogenesis [131,132,133,134]; doubled BDNF levels, increased dendritic length, increased anti-apoptotic Bcl2 and Bcl-XL, and prevented neuronal death from glutamate and pro-apoptotic BAD, BAX, and caspase 3 [135]. It increased the numbers and size of neural mitochondria [136], increased antioxidants [129,136], minimized neurotoxicity from cytochrome c, and promoted mitochondrial biogenesis [137,138]. Lithium benefitted microvascularity by increasing VEGF secretion [139] and BBB integrity [140]. Lithium’s inhibition of GSK-3β reduced the production and activation of pro-inflammatory mediators [141,142,143].

Memantine induced activation of astrocytes [144] and prevented losses of oligodendrocytes [145], synapses and neurons [146,147,148], and endothelial cells [149]. It is an NMDAR antagonist, enters the NMDAR’s Ca^2+^ channel and decreases its permeability, so it prevents neuronal excitotoxicity and death [146]. Memantine provides synaptic protection via several mechanisms: preventing cytotoxicity by blocking inhibition of a guanosine triphosphatase involved in multiple cellular processes, thus protecting against mitochondrial dysfunction causing cell death via cytochrome c release, ROS and peroxide production [150,151,152,153]. Blocking the ion channel of the acetylcholine receptors also prevents neurotoxicity [154]. Memantine benefitted brain endothelial cells and blocked disruption of the BBB [155].

Minocycline prevented oligodendrocyte toxicity caused by the deprivation of oxygen and glucose [156] and microglial-induced apoptosis [157] by minocycline preventing the inhibition of CREB’s up-regulation [158], and caused by Aβ [159]. Synaptic function improved from increased levels of PSD-95 and dendritic spines [160]. It prevented cognitive decline caused by an antagonist of the NMDAR [161]. The neuronal benefit was also from its increased expression of BDNF, CREB, and phospho-CREB [158] and the proliferation of NPCs [162]. Minocycline potentiated neurite outgrowth [163]. In a transgenic mouse model of Down’s syndrome, minocycline prevented the decline of cholinergic neurons [164]. In a list of 1040 drugs that prevent the release of cytochrome c, minocycline was the second most potent [165]; the mechanism involves inhibiting mitochondrial increases in Ca^2+^ concentration plus inhibition of NADH-cytochrome c reductase and cytochrome c oxidase [165,166]. Finally, minocycline prevented microglial activation [167,168].

Pioglitazone, a PPARγ agonist, prevents phosphorylation of JAK-STAT in astrocytes and thereby induces increases in neurons [169,170], oligodendrocytes [119,170], endothelial cells [171] and decreases in microglia [172]. Ciglitazone and curcumin, also PPARγ agonists, were cytoprotective for astrocytes and reversed their decreased expression of PPARγ receptor after exposure to Aβ25-35 [173,174]; and ciglitazone increased the formation of oligodendrocyte progenitors [170]. Pioglitazone rescued the demyelination caused by anti-MOG autoantibody [175]. The promotion of OPC differentiation into mature oligodendrocytes by IL-4 was mediated by PPARγ or curcumin [176,177]. PPARγ agonism increases the neurogenic differentiation gene, NeyroD1 [170] and protects cortical neurons and axons against toxicity induced by NO or KCl-induced toxicity [169,178].

PPARγ also produces endothelial cell proliferation and angiogenesis [171,179]. In microglia, PPARγ agonists inhibit cytokine production by down-regulating proinflammatory genes [180]. In addition, rosiglitazone up-regulated M2 microglia [180].

Piracetam increased the numbers and function of astrocytes [181]. It also increased the function of synapses [182,183,184,185] and neurons [182]: it decreased neurotoxicity from deprivation of oxygen and glucose, hypoperfusion, ethanol feeding or ethanol withdrawal [183,186,187,188]. It also caused longer neurites [182]. The effects of piracetam may derive in part from the restoration of cell membrane fluidity induced by a conformational change in the phospholipids of the liposomal membrane [185].

By decreasing mitochondrial swelling and permeability caused by excessive Ca^2+^ opening the MPTP, piracetam improved mitochondrial membrane potential and ATP levels, shifted the balance of mitochondrial fission or fusion towards fusion, and reversed the adverse effect of pro-oxidants [189]. Additionally, it reversed the cytotoxic effects of p53 and BAX [183].

Riluzole increased both the gene for the excitatory amino acid transporter (EAAT2) and its expression, thus enhancing glutamate uptake by astrocytes and protecting against excitotoxicity of neurons [190,191]. It benefits synapses, as shown by the inhibition of voltage-activated sodium currents, which prevents the reverse operation of the Na^+^/Ca^2+^ exchanger [192], and by increasing LTP [193]. It caused a 40-fold increase in hippocampal BDNF, creating neurogenesis [194] and protecting against neural degeneration caused by ischemia [195]. N-acetylaspartate is exclusively expressed in neurons, was decreased in PTSD (see above) and was increased in the cerebral cortex by exposure to riluzole [196]. In neurons, riluzole also decreased oxidative stress, lipid peroxidation, and ATP depletion [197,198]. Microglial activation was ameliorated by riluzole [199], which also upregulated the mRNA levels of M2 markers (anti-inflammatory) and downregulated those of M1 markers (pro-inflammatory) [200].

## 4. Discussion

This article argues that because in behavioral reactions caused by stressors (the paradigm for which is PTSD), there are such numerous and complex metabolic and genetic abnormalities that addressing the changes that those abnormalities mediate in brain cell types is the optimum approach to formulating a rational pharmacotherapy that may be used to supplement psychotherapy. Data show that those stress reactions are associated with decreases in astrocytes, oligodendrocytes, synapses and neurons, and endothelial cells/cerebral microcirculation, and increases in microglia, particularly M2 microglia that are pro-inflammatory. Drugs that beneficially address the changes in those brain cell types include clemastine, dantrolene, erythropoietin, fingolimod, fluoxetine, lithium, memantine, minocycline, pioglitazone, piracetam, and riluzole.

The cure for PTSD requires addressing the cerebral cell types that are responsible for its pathogenesis. While it is not necessarily the case that, for individual subjects with PTSD, all five cell types are involved, for practical purposes, it is prudent to assume that they are; so, drugs that address all five cell types are the best candidates for a regimen that has the best chance for successful reversal of the behavioral problem caused by stressors. It is recommended to use combinations of two or three drugs in the above list in order to minimize dosages and the risk of both serious adverse reactions (SAE) and drug-drug interactions, including with drugs used for co-morbid conditions. The best chances would come from the use of drugs that address all five cell types. Those are erythropoietin, fluoxetine, lithium, and pioglitazone. However, unless there is a compromise of the blood-brain barrier (BBB), erythropoietin will not cross it. So, the choice of a two-drug combination must be chosen from fluoxetine, lithium and pioglitazone; however, fluoxetine and lithium potentially cause the most SAE. Therefore, a clinical trial to test whether two of the suggested drugs will, in fact, cure PTSD should administer pioglitazone plus either fluoxetine or lithium, using low dosages: pioglitazone 15 mg/day, lithium 75 mg/day, or fluoxetine 10 mg/day. Four cell types benefit from clemastine (which causes no SAE), fingolimod, and memantine; one drug from this group could be added to the two-drug combination to form a three-drug combination. Clinical trials would demonstrate the validity of both the proposed concept and the choice of drugs for treatment and would also show the percentage of patients benefitting from the chosen combination of drugs. It is important to note that this use of the suggested drugs would be off-label and that a clinical trial is required to establish their efficacy for the reversal of PTSD.

Table 1 provides an abbreviated summary of some of the pharmacodynamic and pharmacokinetic issues regarding the suggested drugs. 

It must also be emphasized that urine tests should be obtained in appropriate patients if there is any suspicion of self-treatment by the use of alcohol, cannabis, or illegally-obtained drugs, and if other providers have been involved, they should be contacted for whatever useful information that they had obtained.

In a clinical trial that randomly assigns participants to receive active drugs or a placebo, the calculation of the required number of participants would be based upon the primary objective showing ≥50% more reversal of PTSD in those using active drugs than those participants randomized to take a placebo. The number in each group would be calculated on the basis of an annual drop-out rate of 10%. The planned duration of treatment would be 48 weeks with an option to extend for a further 48 weeks if recommended by the study board (DSMB), which will meet every 24 weeks to review the data. Participants would be aged ≥18, with a diagnosis of PTSD made according to the current diagnostic and statistical manual of mental disorders (DSM). Exclusions would be individuals with other active psychiatric diagnoses, particularly MDD and substance abuse.

## 5. Conclusions

Data show many metabolic and genetic associates of behavioral conditions due to stressors, for which PTSD is paradigmatic. Addressing all of them would require an unacceptable number of drugs. However, the effects of those metabolic and genetic associates affect brain cell types, whose changed functions result in behavioral disturbance;In PTSD, there is an impairment in the numbers and functions of all of the five brain cell types. Therefore, optimum pharmacotherapy for PTSD must benefit astrocytes, oligodendrocytes, neurons, endothelial cells, and microglia;Each of fluoxetine, lithium, and pioglitazone corrects the changed functions of all five cell types. Administering two of them together should benefit all patients with PTSD, including those having psychotherapy. For a two-drug regimen, it is recommended to use low dosages of pioglitazone paired with low doses of either lithium or fluoxetine. One additional drug, taken from among clemastine, fingolimod, and memantine., would form a three-drug regimen;The cure of PTSD by the above groups of drugs in patients with an inadequate response to psychotherapy should be validated by clinical trials.

## Figures and Tables

**Table 1 jcm-12-01680-t001:** Suggested drugs: their dosages, pharmacodynamics, and pharmacokinetics.

Drug	Suggested Dose	Serious Adverse Effects (SAE)	Drug-Drug Interactions	Effects of Comorbidities	Effects of Age	Effects of Pregnancy
clemastine	5.36 mg bid (a)	None				
erythropoietin	10,000 IU IM once/month	SAE only if high dosage.				
fingolimod	0.5 mg/day	(b)				
fluoxetine	10 mg/day		(c), (d).		No effect	
lithium	75 mg/day	(e), (f), (g), (h), (i), (j).		Avoid if GFR < 60 mL/min.		Avoid in pregnancy
memantine	5 mg qd ×7d, then bid (j).	(k).		Avoid if GFR < 60 mL/min.		
minocycline	100 mg/day	(l), (m), (n), (o).				
piracetam	1.2 G/day	No SAE	None			
pioglitazone	15 mg/day	No SAE				
riluzole	100 mg/day	(p).				

See text for descriptions of therapeutic effects. Suggested two-drug combinations are pioglitazone plus fluoxetine or pioglitazone plus lithium. Suggested three-drug combinations would add one drug from clemastine, fingolimod, or memantine. (a) this high dose for a study was approved by FDA. (b) fingolimod may cause transient 2^0^ heart block. (c), fluoxetine + lithium may cause serotonergic syndrome. (d), fluoxetine may increase blood levels of antipsychotic drugs. Lithium may cause (e) hypercalcemia, (f) hypothyroidism, (g) rarely hyperthyroidism, (h) GFR may fall; do not use if GFR < 60 mL/min, (i) ~20% patients In clinical trials, quit Rx due to adverse events, (j) NB suggested very low dosage is unlikely to cause problems. (k) Use a lower dose of memantine in (j) elderly patients. (k) do not use if GFR < 60 mL/min. (l) vestibular symptoms are the commonest side effect but infrequent. (m) systemic lupus erythematosus as SAE is serious but rare; ANA testing before commencing Rx is mandatory, (n) other hypersensitivity syndromes may also occur, (o) cutaneous and dental hyperpigmentation occurs with prolonged Rx, (p) Liver enzymes abnormal in ~10% Rx’ed with riluzole.

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
