# Peer review of "Supplementary Pharmacotherapy for the Behavioral Abnormalities Caused by Stressors in Humans, Focused on Post-Traumatic Stress Disorder (PTSD)"

_jcm, 2023, doi:10.3390/jcm12041680_

Round 1
Reviewer 1 Report
In this review the author presents a deep analysis of possibilities of using pharmacological treatments for stressor-induced genetic and metabolic conditions that results in behavioral abnormalities such as Post-Traumatic Stress Disorder (PTSD) and other psychiatric conditions. It is suggested that the carefully recommended pharmacotherapy can be used as a supplement to psychotherapy
The manuscript provides important and useful clinical information concerning the basis of the behavioral abnormalities and describes the mechanism of action and possible use of a long list of therapeutic drugs. The review is based in a carefully revision of the pertinent literature.
Except for some inadequate points that must be corrected. These are the points:
1. Line 41: Since Ref 1 is published in a referenced journal the existence of a DOI number is obvious. Thus, the complement indication “(ref with DOI)” can be excluded.
2. Lines 248-251: The whole text concerning Renin-Angiotensin system, starting with the role of ACE, the physiological effects of AT1 and AT2 as well the role of their receptors are misconceived and require adequate correction.
Author Response
1. The “ref with DOI” should have been removed with the submission. It is now removed.
2. Lines 248-251. I agree with the comment. I have removed those lines and replaced them; I hope that the replacement is satisfactory. The replacement requires two additional citations but unfortunately, they were given numbers 1 and 2 at the end of the citation list; ENDNOTE could not change that for me, so
the Copy editor will need to replace those extra numbers 1 and 2 with 45A and 45B).
Reviewer 2 Report
Supplementary pharmacotherapy for the behavioral abnormali-2 ties caused by stressors in humans, focused on PTSD
I would like to thank the authors for their great efforts to collect these data about the post-Traumatic Stress Disorder (PTSD). They addressed the cellular abnormalities, the used drugs and the molecular mechanism that these drugs can improve the brain cells.
This manuscript is well written.
Using abbreviations in the title, and the abstract without indicators is not preferable. For example; (PTSD)
Line 41; what is the reference with DOI?
The language needs to be improved especially in the abstract. The sentences need to be more clear.
The title is suitable.
The research is well designed.
The order of the review is good.
I strongly recommend the authors to add some figures to express the idea facilitating the reader to follow the genetic and metabolic pathways
I also strongly recommend the authors to add some tables to collectively show the genetic changes in cells. For example; Oligodendrocytes and neurons with the references which will be summarize the data more efficiently.
The conclusion is very critical and well written.
The references are suitable and updated.
Author Response
1). I have expanded PTSD in the title.
2). I have removed the “ref with DOI” from line 41 (it should not be there!).
3). Since I am retired and without funds, I cannot arrange to have a graphic artist make a figure.
4). The genetic changes are very complex and a Table will not simplify them.
Reviewer 3 Report
This perspective by Jeffrey Fessel is very interesting. Given the metabolic and genetic changes underlying PTSD are numerous, the author state that addressing the cell-types changes during PTSD may be a good strategy. I do not totally agree the view, but I think that this manuscript still is eye-opening. My comments are as following:
1, Maybe, the number changes of cells are rather downstream in the pathology of PTSD. If the players in the upstream are untreated, it is hard to make the down players right.
2, To function properly, not only the cell number is important, but also the right organization of the cells.
3, About the following sentence:
‘Data show that those stress reactions are associated with decreases in astrocytes, oligodendrocytes, synapses and neurons, and endothelial cells/cerebral microcirculation, and increases in microglia, particularly M2 microglia that are pro-inflammatory.’
In general, M1 microglia are pro-inflammatory; M2 are pro-regenerative.
Author Response
1). M2 was a typo error; I have corrected it to M1.
2).Regarding the comment that the brain cell-types could be downstream, I want to remind the Reviewer that the concept proposed in the MS, is to target the brain cell-types instead of targeting the numerous upstream, genetic, metabolic, and environmental factors, which would require an avalanche of drugs. Changes in those downstream cell-types are the ultimate causal factors, so addressing them is simpler, requires only two or three drugs, and should be beneficial.
3). I agree that the organization of the cells is important but that would be addressed by employing medications that consider all of the affected cell-types.
Round 2
Reviewer 2 Report
I would like to thank the authors for their great efforts. Their modifications made the manuscript better.
Meanwhile, still, there is no illustrative figures or tables.
Commertially, such action will decrease the value of the manuscript. It will deal with the visual reader and to easily grasp the important points.
I strongly recommend to add such visual aids to re-evaluate the manuscript.
Author Response
I appreciate your suggestion but cannot comply for two reasons:
- I have no funds to pay for a graphic illustration.
- The genetics are extremely complex and a Table would be equally complex and confusing. The few readers who are Interested, can check the citations for the details.